# Research on the Control of Excavation Deformation of Super Deep Foundation Pit Adjacent to the Existing Old Masonry Structure Building

Huajun Xue [1,2]

1   Inspection and Certification Co., Ltd., MCC, Beijing 100088, China; xuehuajun7792@sina.com
2   Central Research Institute of Building and Construction Co., Ltd., MCC Group, Beijing 100088, China

**Abstract:** In order to ensure the safety and stability of the existing old masonry structure houses in the process of dewatering and excavation of the super deep foundation pit of the subway, the support form of a water stop curtain combined with bored cast-in-place piles and internal support is adopted, and the rotary jet grouting piles are constructed around the houses, and sleeve valve pipes are embedded, and the soil and house foundation are grouted and strengthened. The deformation of the building foundation is analyzed by the finite element method. The results show that the deformation of adjacent buildings is mainly uniform at the initial stage of foundation pit dewatering and excavation. With the increase of foundation pit dewatering and excavation depth, the deformation of adjacent buildings shows significant differential characteristics, and the maximum displacement of buildings is settlement deformation. The field monitoring data show that the actual deformation trend and value range of the building structure are basically consistent with the finite element calculation results, and no new damage is found in the building structure during the construction process. Effective foundation pit support method and soil layer reinforcement method can effectively reduce the impact of foundation pit on the deformation of adjacent buildings.

**Keywords:** super deep foundation pit; old masonry buildings; finite element calculation; soil layer reinforcement; deformation control

## 1. Introduction

In recent years, with the rapid development of China's economy and the accelerating process of urbanization, the scale and population of cities have increased greatly, resulting in the decreasing of ground space resources. Therefore, the construction and research of urban underground space has gradually attracted the attention of civil engineering constructors and scientists [1–4]. Foundation pit excavation is an important part of underground engineering construction. However, with the increasing excavation depth and excavation scope of foundation pit, it will inevitably affect the stability and safety of adjacent buildings [5,6]. Therefore, it is particularly necessary to study the influence of deep foundation pit excavation on the deformation and structural safety of adjacent buildings.

At present, many scholars have studied the impact of foundation pit excavation on surrounding buildings. Frischman [7] studied the causes of building damage caused by foundation pit excavation and put forward corresponding protective measures. Xu et al. [8] analyzed the influence of excavation of deep foundation pit on the surrounding environment. Li et al. [9] analyzed the influence of deep foundation pit excavation in a subway station on adjacent bridge piles. Li and Xi [10] and Shi et al. [11] studied the influence of deep foundation pit excavation on the settlement of surrounding buildings by finite element method. Gong et al. [12] studied the deformation law of adjacent buildings under the influence of the superposition of foundation pit and lower tunnel group construction. Wei [13] used finite element method to analyze the influence of foundation pit distance, foundation pit type, excavation depth, precipitation, depth of diaphragm wall and foundation pit

excavation method on the deformation of adjacent frame structure. Wang et al. [14] studied that the five-sided water stop structure formed by rotary jet grouting piles around the pit and at the bottom of the pit can effectively reduce the consolidation settlement of deep foundation excavation on the surrounding existing railway line subgrade. Yu et al. [15] analyzed the influence rule of external deep foundation pit excavation on the deformation of the metro line in operation. These studies have analyzed the influence of deep foundation pit excavation on the surrounding environment and buildings. However, due to the complexity and variety of construction environment and construction methods, the understanding of the deformation law of adjacent buildings caused by deep foundation pit excavation is not clear enough, and further research is needed.

Based on the deep foundation pit construction of Renmin South Road subway station in Taiyuan, in this paper, Midas software 2022 is used to simulate the deformation of adjacent buildings during the dewatering and excavation of the deep foundation pit of the subway, and combined with the on-site monitoring results, the displacement law of adjacent buildings under the influence of the dewatering and excavation of the deep foundation pit is analyzed.

## 2. Engineering Background

### 2.1. Engineering Situation

Renmin South Road Station is located at the intersection of Renmin South Road and Guihua No. 4 Road in Xiaodian District, Taiyuan, and is arranged along Renmin South Road from north to south. The underground structure of the station is two floors, two columns and three spans, and the surrounding area of the station is relatively wide. The station is constructed by open-cut method, and the total length of the station is 307.55 m. The standard height of the main station is 15.52 m, and the width is 21.6 m. The thickness of the roof covering the center of the effective platform of the station is about 3.52 m, and the buried depth of the structural floor is about 19.04 m. The average depth of the station foundation pit is 20 m, and the width is 21.6 m, which belongs to the super large deep foundation pit. The excavation shall be carried out in layers, sections and blocks.

There is a residential building in the Xiqiaoyuan community on the northeast side of the station. The residential building was built in the 1980s, and its structural form is a brick–concrete structure house with six floors above the ground and one floor below the ground. The floor and roof panels are prefabricated houses, belonging to a typical old masonry structure house. The house is 49 m in the east–west direction and 17 m in the north–south direction, with a construction area of 5194.45 m$^2$. The nearest distance between the foundation pit of Renmin South Road Station and the edge of Xiqiaoyuan Community nearby is 6.9 m, and the included angle between the foundation pit and the retaining wall is 60. Foundation pits are excavated immediately adjacent to the structure of the house. Therefore, it is necessary to study the impact of foundation pit excavation on the safety of residential structures. The relative positional relationship between the station foundation pit and the residential building in this community is shown in Figure 1.

### 2.2. Engineering Geology and Hydrogeology

See Table 1 for the physical and mechanical properties of the soil layers in the station site from top to bottom.

The groundwater in this area is shallow phreatic water in Quaternary loose layer. The buried depth of groundwater is 1.6–4.3 m; the thickness of shallow phreatic aquifer is about 30 m; the water level varies by about 1–2 m; and the water inflow of a single well is 100–300 m$^3$/d. Shallow diving is easily affected by atmospheric precipitation and human activities because of its shallow depth and small amount of water.

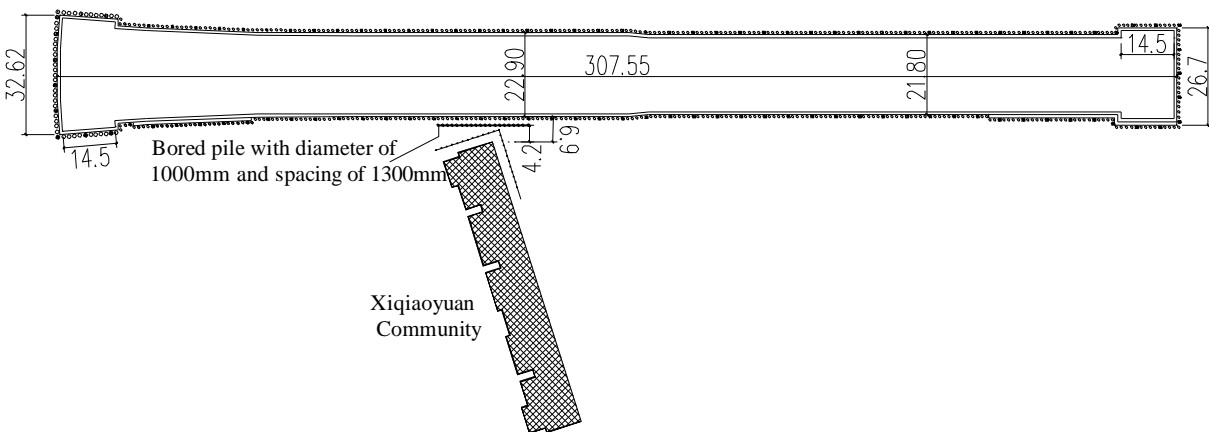

**Figure 1.** Relative location relationship between station foundation pit and nearby residential buildings, special reinforcement measures (Unit: m).

**Table 1.** Physical and mechanical parameters of materials.

| Soil Layers | Average Thickness (m) | Soil Natural Density (g/cm$^3$) | Young's Modulus (MPa) | Cohesive Strength (kPa) | Internal Friction Angle (°) |
|---|---|---|---|---|---|
| Miscellaneous fill | 2.37 | 1.76 | 8.31 | 8.0 | 10.0 |
| Silty clay 1 | 1.50 | 1.93 | 18.90 | 18.1 | 19.0 |
| Clay silt 1 | 4.05 | 1.96 | 24.00 | 12.9 | 20.5 |
| Fine sand 1 | 3.16 | 2.00 | 31.20 | 3.1 | 21.5 |
| Fine sand 2 | 2.64 | 2.02 | 24.50 | 3.7 | 22.5 |
| Medium sand | 2.68 | 2.04 | 40.08 | 1.5 | 24.0 |
| Silty clay 2 | 1.91 | 1.95 | 27.09 | 18.8 | 20.0 |
| Clay silt 2 | 9.80 | 2.01 | 34.95 | 14.8 | 21.5 |
| Silty sand 3 | 2.70 | 2.02 | 40.65 | 3.2 | 23.5 |

*2.3. The Retaining Structure of Foundation Pit*

The main retaining structure of the station is supported by bored piles with steel support. The diameter of bored piles is 1000 mm, and the spacing is 1300 mm. The steel support system adopts the internal support method of φ809 steel pipe and reinforced concrete support, and the thickness of the supporting steel pipe is 16 mm. Considering the influence of groundwater, a single-row triaxial deep mixing pile is used to construct the waterproof curtain, with a pile diameter of 850 mm and a spacing of 600 mm.

Steel support erection and earthwork excavation of deep foundation pit are two key processes which are closely related to deep foundation pit construction. The timing and position of steel support erection are directly related to the stability of deep foundation pit. In the process of foundation pit excavation, three steel supports are set from the top of the crown beam to the bottom of the foundation pit. When the first steel support is excavated, the first concrete shall be constructed; then, the first concrete shall be excavated downwards in turn; the second and third steel supports shall be erected in turn; and finally the foundation shall be excavated. Figure 2 shows the supporting structure diagram of the main structure of the station foundation pit.

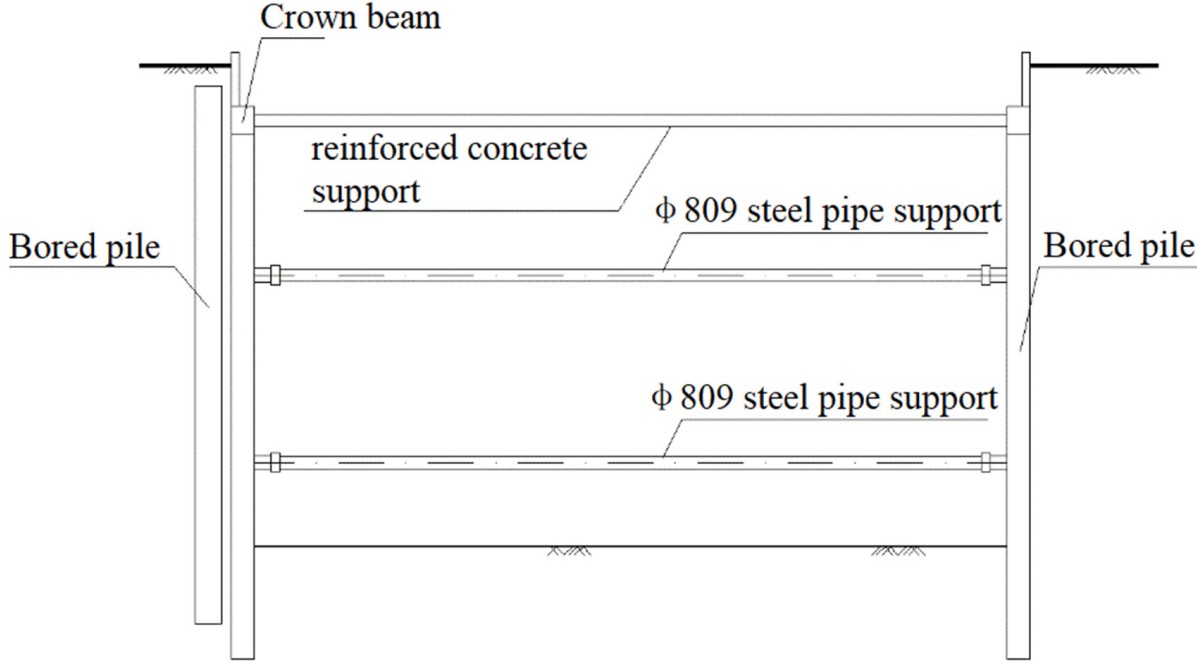

**Figure 2.** Schematic diagram of the envelope structure of the main structure of the station foundation pit.

### 3. Numerical Calculation and Deformation Control and Countermeasures

In order to effectively control the impact of foundation pit dewatering and excavation on the deformation of adjacent buildings, rotary jet grouting piles are constructed around the buildings, and sleeve valve pipes are embedded to reinforce the soil and building foundation, as shown in Figure 1.

#### 3.1. Calculation Model

In this paper, Midas GTS NX finite element software 2022 is used to analyze the influence of subway super deep foundation pit excavation on adjacent buildings. Midas GTSNX is a general finite element analysis software developed for the geotechnical field, which is widely used to study the stress and deformation of structures in the construction process of underground works and foundation pits. As shown in Figure 3, there is an established three-dimensional numerical calculation model. The model has a length of 224 m, a width of 466 m and a thickness of 60 m. According to the engineering geological conditions of the foundation pit site, a total of nine rock and soil layers are established in the model. See Table 1 for the thickness and physical and mechanical parameters of each soil layer. The Mohr–Coulomb constitutive model is adopted for the constitutive relationship of soil layer, and the elastic constitutive model is adopted for the building foundation, concrete and reinforcement. The model limits the horizontal movement of the x and y axes, and the bottom of the model moves in all directions. The upper part of the model is a free surface. This model focuses on the influence of foundation pit excavation on the settlement and horizontal displacement of adjacent residential buildings. Therefore, in order to simplify the calculation, the load generated by the residential building in the nearby Xiqiaoyuan community is converted into plane uniform load which acts on the soil layer in the model.

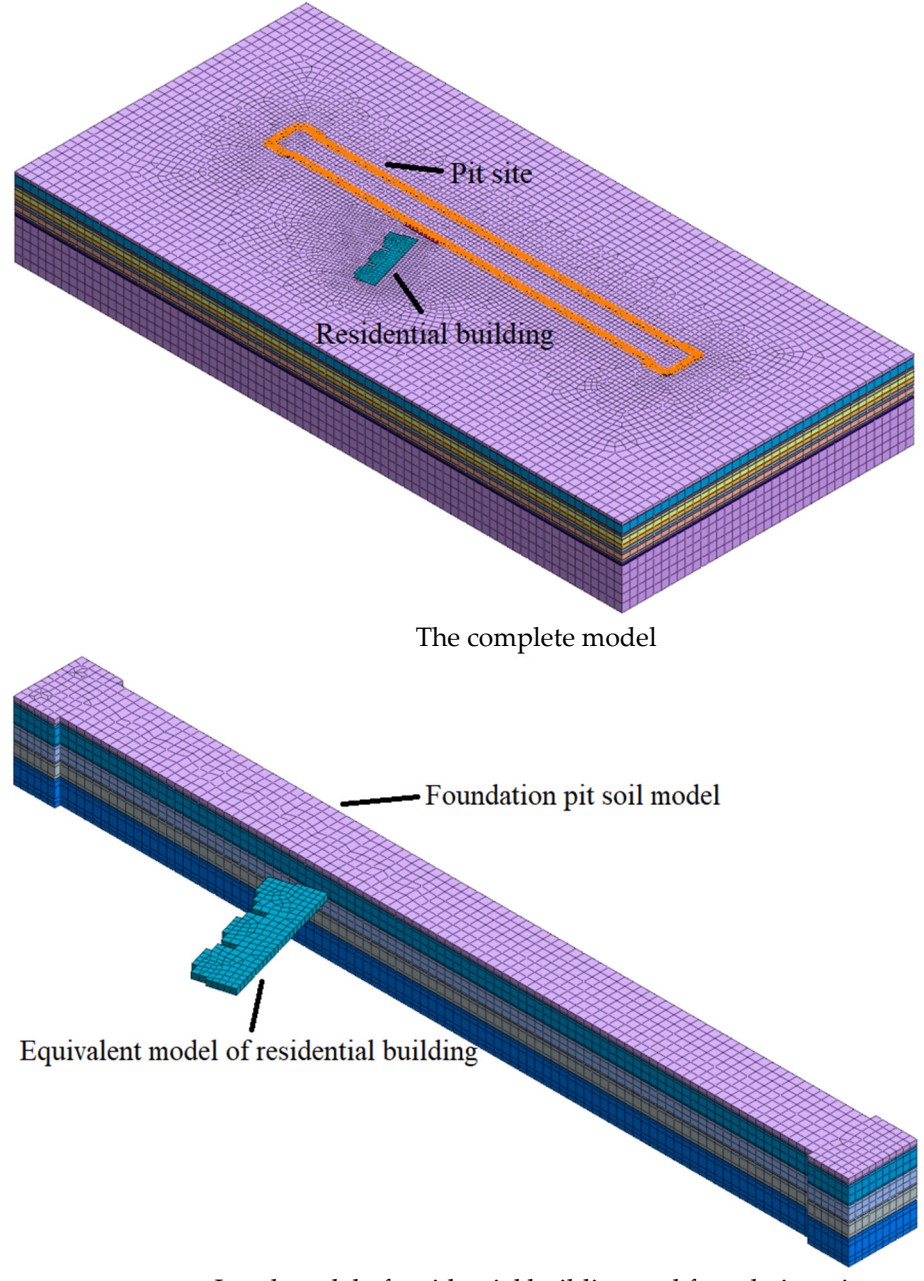

The complete model

Foundation pit soil model

Equivalent model of residential building

Local model of residential building and foundation pit

**Figure 3.** The numerical calculation model.

*3.2. Model Calculation*

According to the actual construction process of deep foundation pit excavation in this subway station, considering the actual construction steps such as layered excavation, excavation and support, the whole excavation process is divided into five working conditions, as shown in Table 2. Figure 4 shows the calculation model of each working condition.

**Table 2.** Division of precipitation and excavation conditions in foundation pits.

| Working Condition Sequence | Excavation Operation | Support Operation |
| --- | --- | --- |
| 1st working condition | Construct support structure before excavation of foundation pit | Construct retaining pile, waterproof curtain, lattice column, soil construction, rotary blast pile and embedded sleeve valve pipe grouting reinforcement |
| 2nd working condition | Precipitation to −3.0 m in the foundation pit and −2.0 m in excavation | Construct the crown beam and concrete support at the top of the pile |
| 3rd working condition | Precipitation to −8.5 m in the foundation pit and −7.5 m in excavation | Construct the second steel support |
| 4th working condition | Precipitation to −15.5 m in the foundation pit and −14.5 m in excavation | Construct the third steel support |
| 5th working condition | Precipitation to −20.0 m in the foundation pit and −19 m in excavation | |

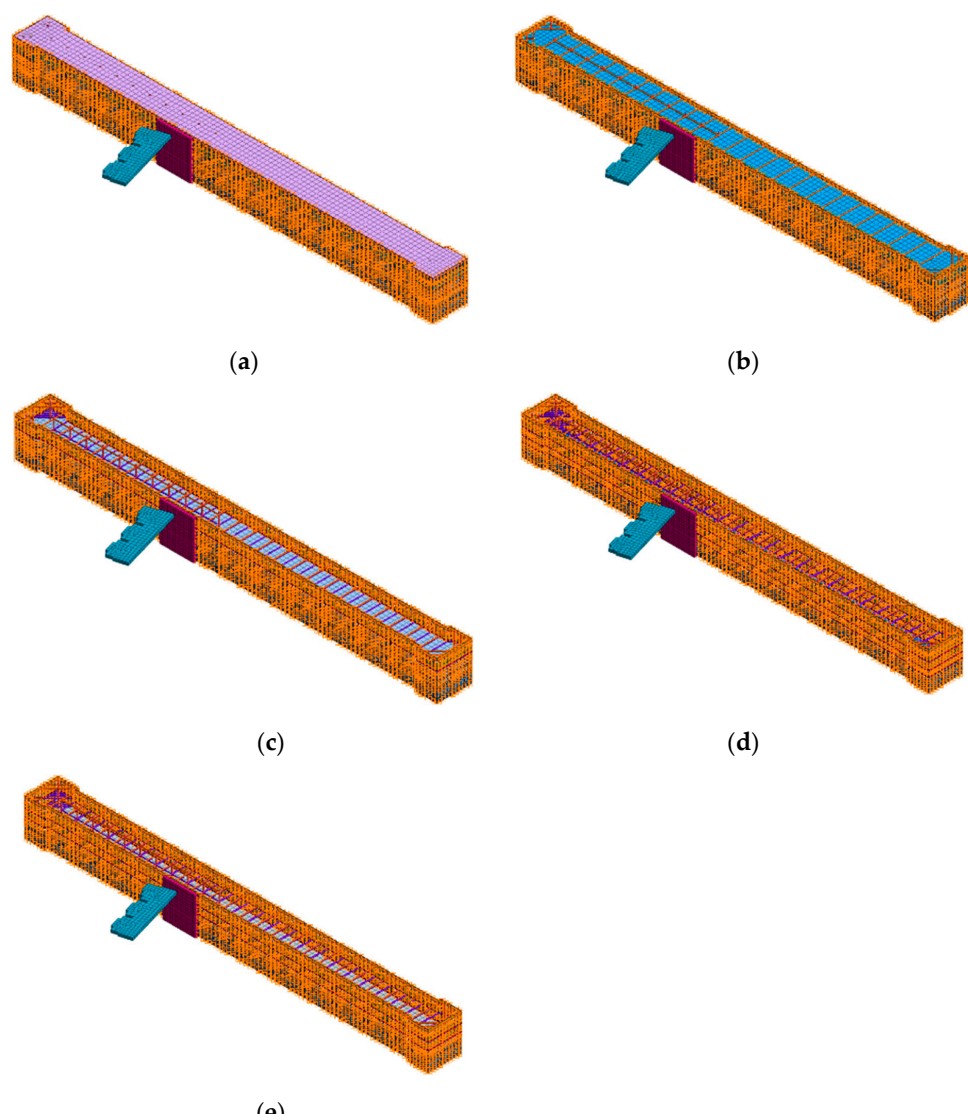

(a)　　　　　　　　　　　　　　　　　(b)

(c)　　　　　　　　　　　　　　　　　(d)

(e)

**Figure 4.** Numerical calculation model of precipitation and excavation conditions of station foundation pit. (**a**) 1st working condition; (**b**) 2nd working condition; (**c**) 3rd working condition; (**d**) 4th working condition; (**e**) 5th working condition.

## 4. Results and Analysis

### 4.1. Settlement Deformation

Figure 5 shows the cloud picture of settlement and deformation of residential building under various working conditions with the excavation of foundation pit. As can be seen from the figure, after the construction of foundation pit retaining piles, waterproof curtain, lattice columns, etc., the settlement of residential buildings is less than 4.0 mm, and the change of settlement is not obvious. However, with the increase of the excavation depth of deep foundation pit in subway station, the settlement of the ground surface around residential buildings is increasing. This is mainly due to the settlement of adjacent residential buildings caused by the excavation of foundation pit, soil unloading and deformation of the supporting structure. In addition, the settlement of residential buildings also shows significant differentiation characteristics, and the settlement near the foundation pit excavation side is greater than that far from the foundation pit. Additionally, with the excavation of foundation pit, the phenomenon of differential settlement of residential buildings becomes more obvious. When excavated to the base, the maximum settlement of the residential building is 19.34 mm, which is located on the side closest to the foundation pit excavation. The minimum settlement value of residential building is 12.54 mm, which is located on the side away from the excavation of foundation pit.

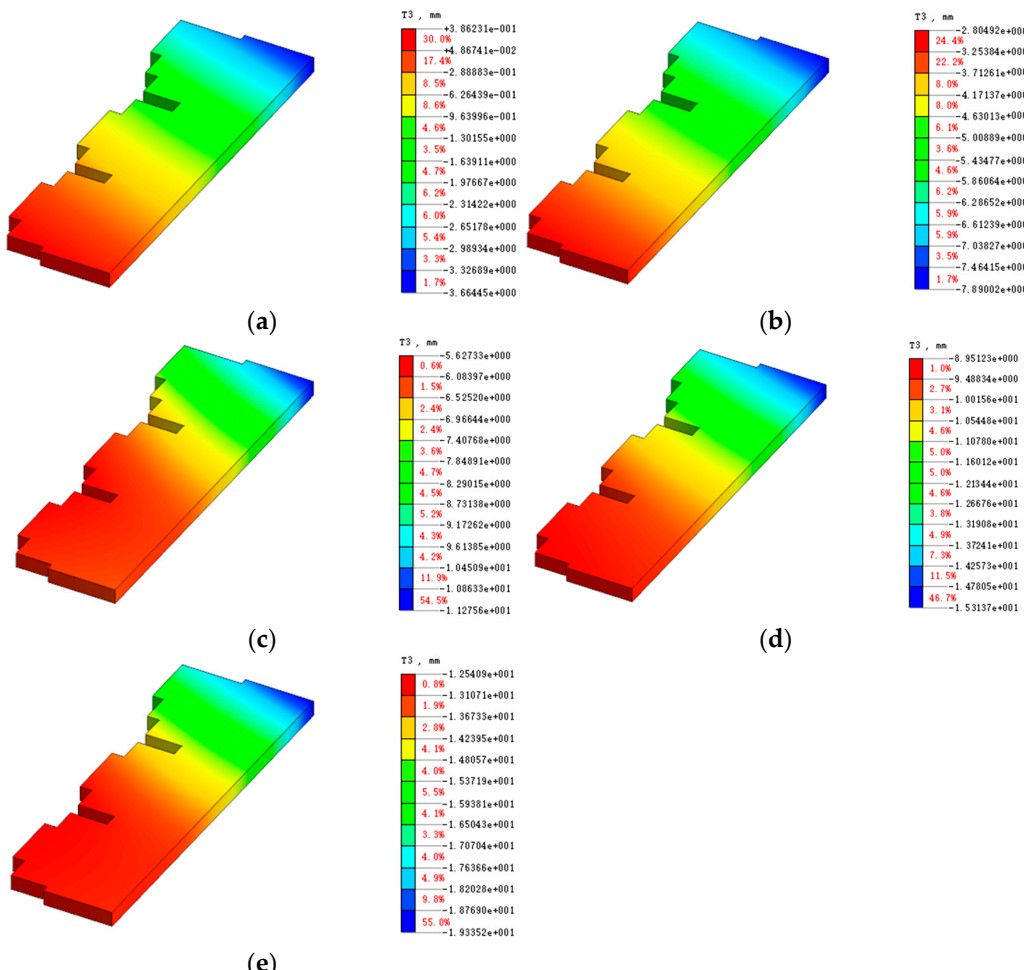

**Figure 5.** Cloud diagram of settlement deformation of residential buildings in the excavation stage under different working conditions of foundation pit. (**a**) 1st working condition; (**b**) 2nd working condition; (**c**) 3rd working condition; (**d**) 4th working condition; (**e**) 5th working condition.

Figure 6 shows the curve of the maximum vertical settlement of the building and the change of its settlement speed with the excavation depth of the foundation pit. With the excavation of foundation pit, the settlement speed of residential buildings is gradually decreasing. In the early stage of foundation pit excavation, the settlement rate of residential buildings is the highest because there is no steel support. With the excavation of foundation pit, the settlement speed of residential buildings gradually decreases, which indicates that the settlement of residential buildings gradually tends to be stable. This shows that the steel support can inhibit the lateral expansion of soil after foundation pit excavation, effectively slowing down the settlement of residential buildings, thus slowing down the further occurrence of deformation, and providing security for the safety and stability of residential buildings.

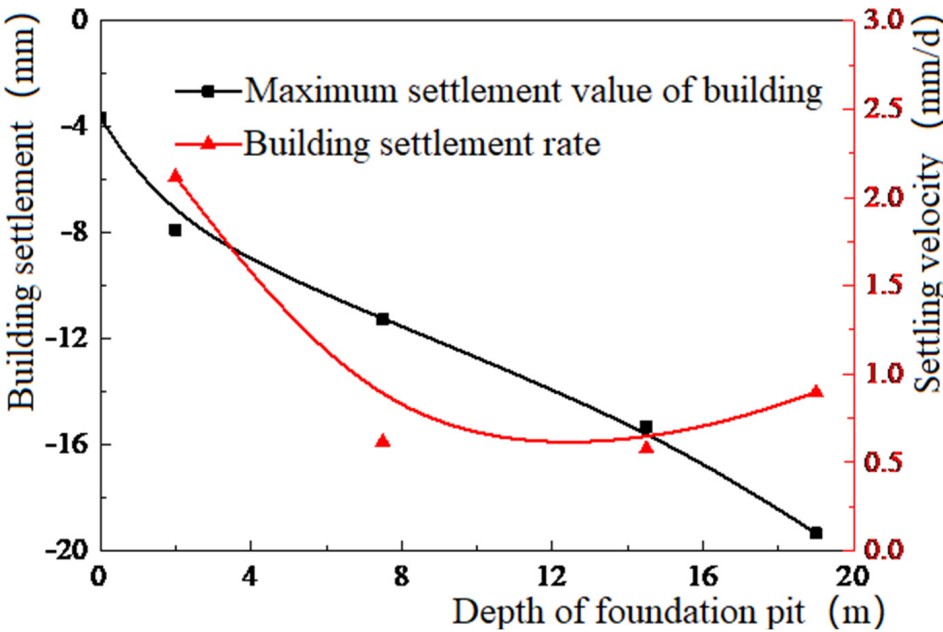

**Figure 6.** The settlement deformation and velocity of residential buildings change with the depth of foundation pit excavation.

### 4.2. Displacement and Deformation Analysis

Figures 7 and 8 are the contour of horizontal displacement along the short side direction and the contour of horizontal displacement along the long side direction of residential buildings under different working conditions, respectively. Because the station foundation pit is excavated near the existing Xiqiaoyuan community, the horizontal displacement of residential buildings is mainly along the long side direction. Similar to the change of settlement, the horizontal displacement of residential buildings also shows the characteristics of differential change.

Figure 9 shows the variation curve of horizontal displacement of residential building with foundation pit excavation. During the excavation of the subway station foundation pit, the horizontal displacement of the residential building along the short side gradually tends to be stable, and its maximum value is 2.34 mm. However, the horizontal displacement along the long side increases gradually with the increase of the excavation depth, and its maximum value is 5.05 mm, which is about twice the horizontal displacement along the short side. This may be because the horizontal displacement along the long side direction is perpendicular to the free surface of the foundation pit, and the building tends to tilt along this direction, resulting in the increase of horizontal displacement in this direction. Therefore, in the process of foundation pit excavation, special attention should be paid to the development of horizontal displacement of buildings in the direction perpendicular to the free surface of foundation pit, and the horizontal displacement of buildings in this direction should be monitored in time to ensure the safety of buildings. In addition, by

comparing the settlement and horizontal displacement of residential buildings, it can be found that with the excavation of foundation pit, the maximum horizontal displacement of adjacent buildings is about 0.25 times of the maximum settlement.

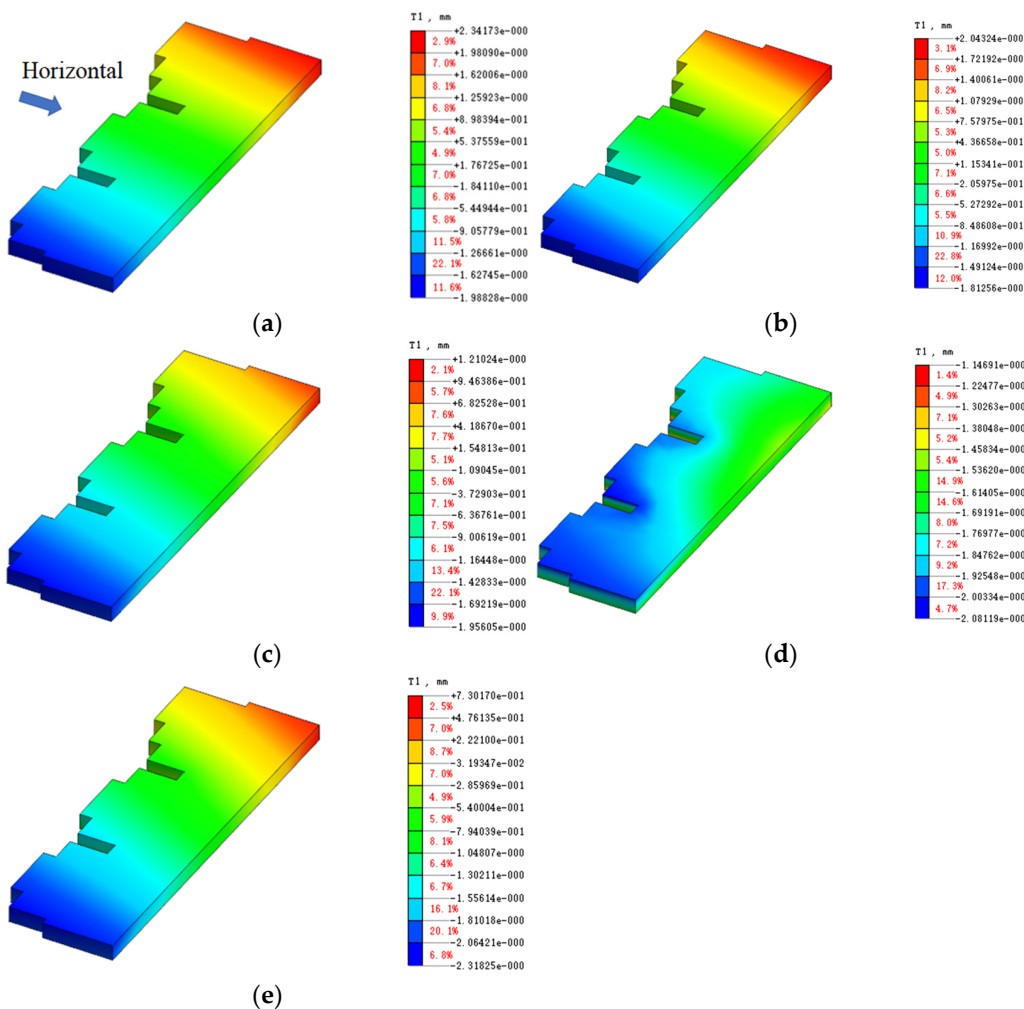

**Figure 7.** Cloud images of horizontal displacement of buildings under different working conditions. (**a**) 1st working condition; (**b**) 2nd working condition; (**c**) 3rd working condition; (**d**) 4th working condition; (**e**) 5th working condition.

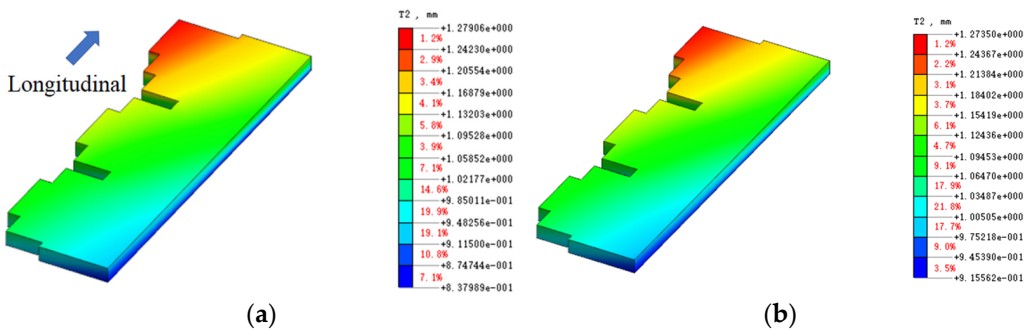

**Figure 8.** *Cont.*

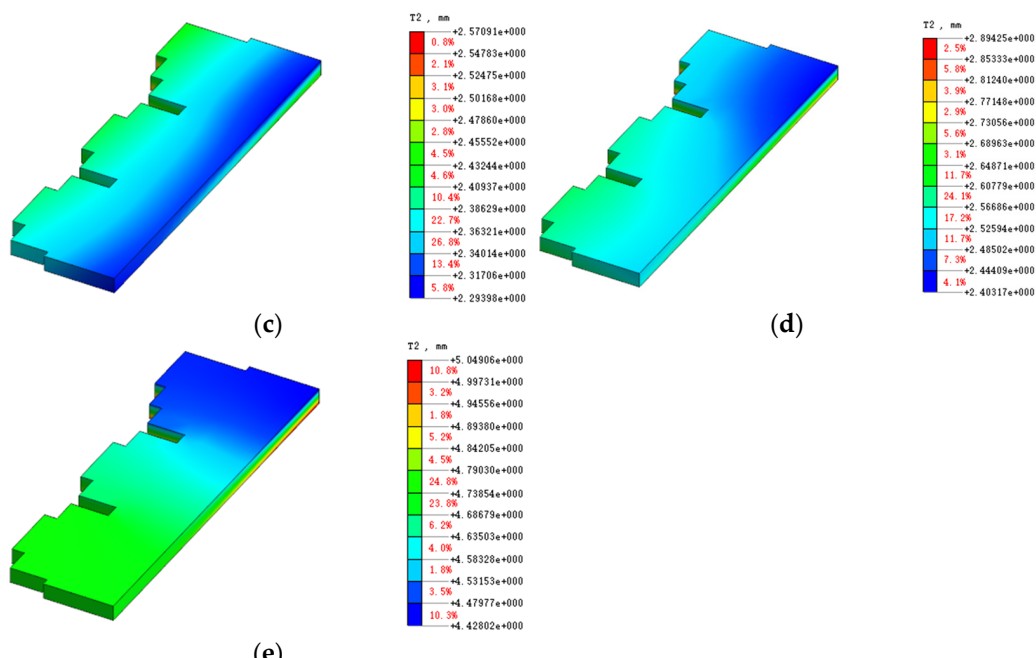

**Figure 8.** Cloud images of longitudinal displacement of buildings under different working conditions. (**a**) 1st working condition; (**b**) 2nd working condition; (**c**) 3rd working condition; (**d**) 4th working condition; (**e**) 5th working condition.

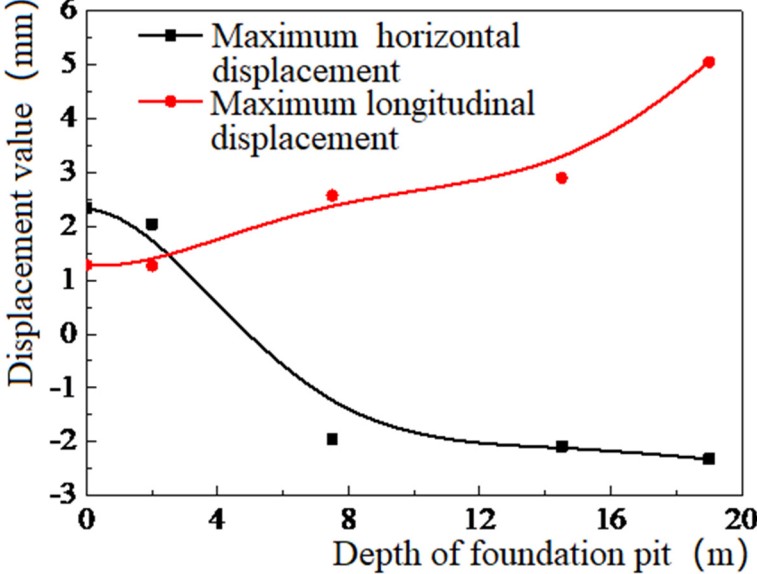

**Figure 9.** Horizontal level of residential buildings changes with the excavation depth of foundation pits.

## 5. Analysis and Countermeasures of Reinforcement and Support Effect on Site

In the process of foundation pit excavation, monitoring points are set around residential buildings, as shown in Figure 10. The settlement of each monitoring point is recorded. Two monitoring points are selected at one side near the foundation pit and one side away from the foundation pit around the residential building, respectively. At the same time, the simulation calculation results of two monitoring points on the side near the foundation pit are selected, and the variation curve of the simulation calculation and the actual monitoring settlement with the excavation depth is shown in Figure 11. As can be seen from the figure, in the early stage of foundation pit excavation, the vertical displacement of each

monitoring point basically changes all the time, which shows that the residential building is characterized by approximately uniform subsidence at this time. With the increase of excavation depth of the foundation pit, the vertical displacement of different monitoring points shows certain differences. The vertical displacement of the monitoring point near the foundation pit increases significantly, while the vertical displacement of the monitoring point far from the foundation pit increases slowly. At this time, the building shows the characteristics of differential settlement. With the continuous excavation of the foundation pit, the change of the vertical displacement of the monitoring points slows down again, and the vertical displacement of each monitoring point gradually tends to be stable. It shows that the supporting structure effectively inhibits the deformation of the surrounding soil layer. In the deformation process of the monitoring point, the vertical displacement of JGC-XQY-46 is the largest, because the monitoring point is closest to the foundation pit, and the excavation of the foundation pit has the most significant influence on it, which is also consistent with the numerical simulation results, which verifies the reliability of the numerical simulation. After the foundation pit excavation is completed, the maximum vertical displacement of the building is 23.2 mm, which is 30 mm below the specified limit, meeting the construction requirements. From the data of JGC-XQY-41 and JGC-XQY-46 in Figure 11, it can be seen that the variation trend of the simulated settlement of residential buildings and the actual monitoring settlement with the excavation depth of the foundation pit is basically consistent, and no new damage to the building structure is found during the construction process.

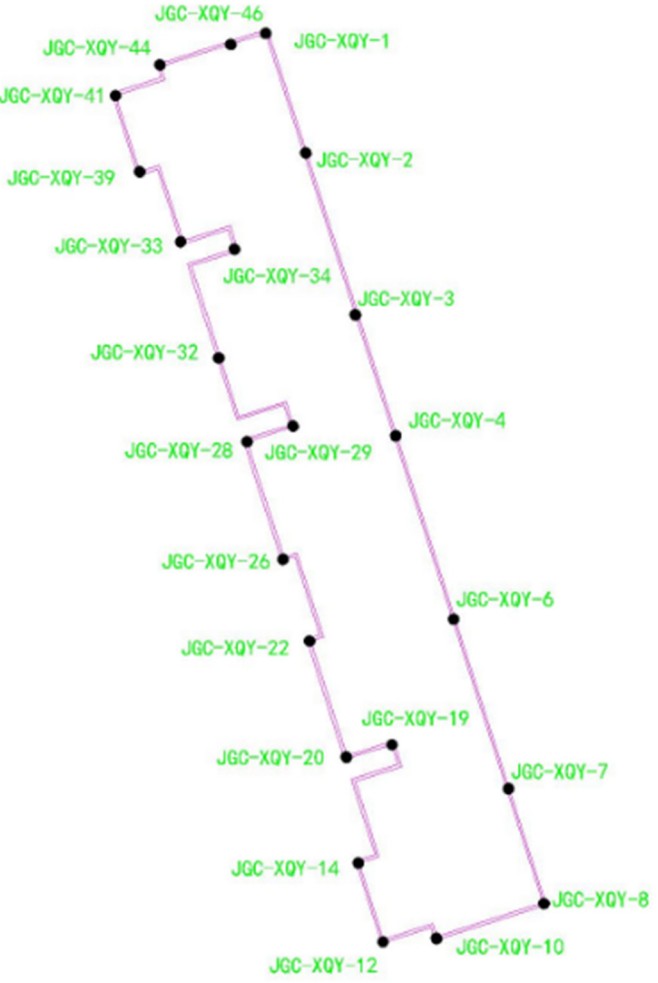

**Figure 10.** The distribution of monitoring points in residential building.

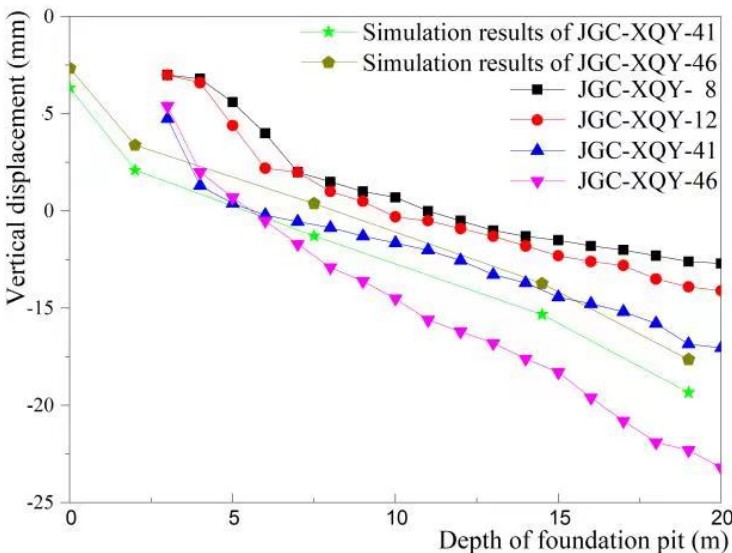

**Figure 11.** The change curve of residential building simulation calculation and actual monitoring.

Since the residential building is close to the foundation pit, the structural safety of the residential building has been tested and identified to ensure the structural safety of the residential building. After the completion of the foundation pit excavation, no new structural damage was found in the residential building. Through the modeling calculation (Figure 12) and evaluation, the risk level of the building was Class B, basically meeting the requirements for safe use of the building.

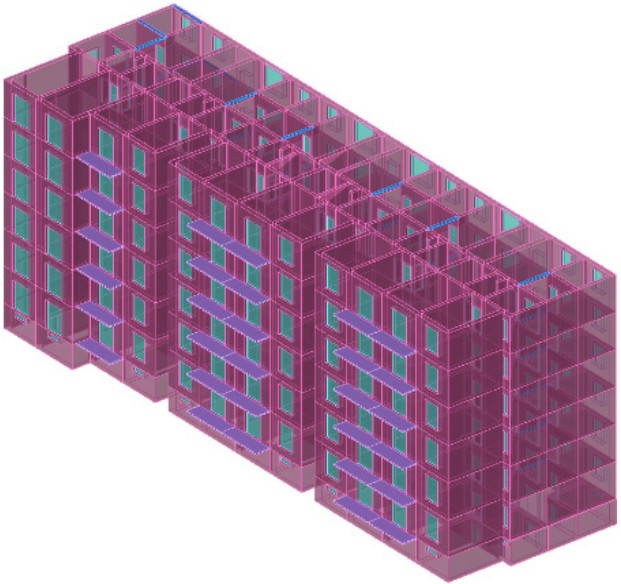

**Figure 12.** Structural model of building safety appraisal.

## 6. Conclusions

Based on the numerical analysis of deep foundation pit excavation in subway station, this paper summarizes the influence of deep foundation pit excavation on the settlement of surrounding buildings and the deformation changes. The main conclusions are as follows:

(1) At the initial stage of foundation pit dewatering and excavation, the deformation of adjacent buildings is mainly uniform. With the increase of foundation pit dewatering and excavation depth, the deformation of adjacent buildings first increased and then gradually stabilized. However, the deformation of houses in different directions shows significant

differences. The maximum displacement of houses is settlement deformation, and the maximum settlement value is four times of horizontal deformation.

(2) The construction monitoring data show that the actual deformation trend and value range of the building structure are basically consistent with the finite element calculation results, and no new damage of the building structure is found during the construction process. The support form of the water stop curtain is combined with bored cast-in-place pile and internal support, and the construction of rotary jet grouting pile around the house and the embedded sleeve valve pipe grouting to strengthen the soil and house foundation can effectively reduce the impact of foundation pit on the deformation of adjacent houses.

(3) The article adopts certain assumptions and simplifications when evaluating and analyzing the deformation impact of subway station foundation pit excavation on adjacent buildings, and the software simulation cannot consider the additional effects generated by other construction before this evaluation stage. Although the calculation results can reflect the actual deformation trend of the existing residential building structure in Xiqiaoyuan Community during the construction of the deep foundation pit structure of the proposed Renmin South Road subway station in Taiyuan City, due to the actual building structure, the complexity and uncertainty of the surrounding environment and conditions make it difficult for the built model to fully match the actual engineering situation, and using the maximum limit allowed by regulations as the deformation control value for existing building structures during the construction process carries certain risks. In the later stage, when evaluating and controlling the deformation of existing building structures in similar projects, three-dimensional scanning technology will be used to measure the overall deformation of the building before subway construction, and the structural deformation will be applied as a forced displacement at the corresponding position of the structural foundation. The internal force of the structure will be calculated and analyzed, and more scientific deformation control indicators for existing building structures will be proposed to reduce construction safety risks within a controllable range.

**Funding:** This research was funded by the Research and development projects in key areas of Guangdong Province (2019B111105002).

**Data Availability Statement:** The data presented in this study are available on request from the corresponding author.

**Conflicts of Interest:** The authors declare no conflict of interest.

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
