# Peer review of "Research on the Control of Excavation Deformation of Super Deep Foundation Pit Adjacent to the Existing Old Masonry Structure Building"

_sustainability, doi:10.3390/su15097697_

Round 1

Reviewer 1 Report

Reviewer’s comments

1.       In the abstract, Lines 8, 14 & 19: the word "set-tlement", found-ation & pro-vides respectively should be written as a single words.

2.       What exactly do you mean by "settlement" and  structural safety? Is it equivalent to vertical subsidence and safety factor respectively?

3.       Missing dimensional units on Fig. 1?

4.       Table 1 has two soil layers referred to as clay silt, each having distinct geomechanical properties. Why? Is it advised to change the names of Clay Silt I and Clay Silt II?

5.       Cite Table 1.

6.       What are the boundary conditions (BCs) of the model?

7.       Is water considered in the modelling calculations?

8.       It is suggested that both Table 2 and Fig. 4 read as the foundation pit excavation stages during numerical modelling calculations

9.       What do the Settlement values in Fig. 5 represent?

10.   The settlement values shown in Fig. 5 are unclear.

11.   The caption of Fig. 6 should read as the developments (e.g., magnitude and rate) of residential building settlements at various excavation depths.

12.    Lines 161–166: Fig. 6 indicates an increase in the magnitude and rate of settlements of the adjacent buildings (e.g., Xiqiaoyuan community) as the depth of the foundation pit excavation advances. But beyond the depth of the last excavation, the rate of vertical settling slows down (e.g., 19 m) due to steel support installation.

13.   The caption for Figures 7 and 8 should be altered to read, respectively, Lateral displacements along the short and long side direction of residential construction at various stages of foundation pit excavations.

14.   It is recommended that Fig. 9 illustrates the relationship between the rate and magnitude of horizontal displacement at different foundation pit excavation along short and long-side directions of surrounding structures.

15.   In Fig. 10, there are a lot of overcrowded monitored points. Using several colours could be beneficial.

16.   Why are there only 4 monitoring points shown in the Fig. 11 legend?

17.   Table 3: what are the  MU7.5 & M2.5 stand for?

Author Response

upload it as a Word file

Reviewer 2 Report

The paper presents a numerical model studying the influence of pit excavation on nearby building foundations. While there is interesting and original analysis and results, there are several shortcomings in the manuscript. Hence, major revision with a fresh round of review is recommended, based on the following specific comments:

1.      Pit excavation affects the foundations of all types of nearby structures, not just buildings. Hence, the title as well as the study aspects should be modified. The Abstract should be concise to highlight the main subject matter and aspect of the paper with brief findings.

2.      The literature review portion lacks from completeness, as numerous important and recent relevant contributions are missing, for example:

https://doi.org/10.1155/2021/6638868

DOI: 10.1088/1755-1315/768/1/012101

https://doi.org/10.3389/feart.2021.735315

3.      The numerical model has not been sufficiently validated with available experimental and field test data. This is the greatest drawback of the manuscript.    

4.      Few mathematical equations and correlations would enhance the quality of the work.

5.      It is not clear whether and how the nonlinear stress-strain response of soil has been incorporated in the model.

6.      The knowledge gap with literature is not clear. The motivation, objective and research methodology is not sufficient.

7.      The significance and practical application of the work requires to be highlighted in a separate heading, preferably with appropriate design recommendations.     

8.      The conclusion should be concise with focus on the primary research findings.

9.      Minor comments:

(a)    Improvement in English write-up is required.

(b)   Some Figures need to be improved, for example: 5, 7, 8.  

Author Response

upload it as a Word file.

Round 2

Reviewer 2 Report

The manuscript is fine now.